# The Structure and Organizations of ICHD-3 Differential Diagnoses through DiffNet: A Pilot Study

**DOI:** 10.3390/diagnostics12112589

**Published:** 2022-10-25

**Authors:** Pengfei Zhang

**Affiliations:** Department of Neurology, Robert Wood Johnson University Hospital, New Brunswick, NJ 08901, USA; pz124@rwjms.rutgers.edu

**Keywords:** headache classifications, order theory, category theory, differential diagnosis, primary headaches, secondary headaches, disease classification, digital health, App application, ICHD-3, diagnostic criteria

## Abstract

Differential diagnosis is fundamental to medicine. Using DiffNet, a differential diagnosis generator, as a model, we studied the structure and organization of how collections of diagnoses (i.e., sets of diagnoses) are related in the ICHD3. Furthermore, we explored the clinical and theoretical implication of these answers. Methods: DiffNet is a freely distributed differential diagnosis generator for headaches using graph theoretical properties of ICHD3: (1) DiffNet considers each ICHD3 diagnosis as a node. (2) An edge exists between two ICHD3 diagnoses if they are connected by either classification hierarchy or are cross-referenced in ICHD3 comment section. In the current project, we generated a set of differential diagnoses using DiffNet for each ICHD3 diagnosis. We then determined algorithmically the set/subset relationship between these sets. We also determined the smallest list of ICHD3 diagnosis whose differential diagnoses would encompass the totality of ICHD3 diagnoses. Results: All ICHD3 diagnoses can be represented by a minimum of 92 differential diagnosis sets. Differential diagnosis sets for 10 of the 14 first digit subcategories of ICHD3 are represented by more than one differential diagnosis sets. Fifty-one of the 93 differential diagnosis sets contain multiple subset relationships; the remaining 42 do not enter into any set/subset relationship with other differential diagnosis sets. Finally, we included a hierarchical presentation of differential diagnosis sets in ICHD3 according to DiffNet. Conclusion: We propose a way of interpreting headache differential diagnoses as partial ordered sets (i.e., poset). For clinicians, fluency in the 93 diagnoses and their differential, as put forth here, implies a complete description of ICHD3. On a theoretical level, interpreting ICHD3 differential diagnosis as poset allows for researchers to translate differential diagnoses sets topologically, algebraically, and categorically.

## 1. Introduction

Differential diagnosis is fundamental to clinical practice; in headache medicine, differential diagnosis is crucial to both workup and treatments [1,2,3]. Therefore, understanding how differential diagnosis is structured and organized is vital for headache specialists.

Since candidate diagnoses in a differential form a collection of objects, this collection satisfies the mathematical definition of a set. To our knowledge, a project studying differential diagnosis hierarchically using set/subset relationship between sets of differentials has not been completed. We sought to explore the precise structure and organization of this hierarchy in headaches.

Although the International Classification of Headache Disorders (ICHD3) provides structure for headache classifications, it does not inherently classify differential diagnoses. Nevertheless, the ICHD3 do provide differential diagnosis considerations by cross-references of diagnosis codes. For example, “headaches secondary to TIA” is not hierarchically associated with migraine with aura but the latter is a differential diagnosis of the former; this can be seen from the observation that migraine with aura (ICHD code 1.2) is listed as a reference in the comment section of headache caused by TIA (6.1.2). We took advantage of this observation in our previous work to construct DiffNet, a freely distributed differential diagnosis generator published under Rutgers University [4,5]. DiffNet is an image of ICHD3 in a graph theoretic form where cross references in ICHD3 are interpreted as “differentials”.

Using DiffNet as a model, we asked: What are the set/subset relationships between differential diagnoses? What is the minimum sets of differential diagnosis sets needed to cover all of ICHD3 diagnosis codes? In answering these questions, we sought to provide a hierarchical classification and ordering of sets of differential diagnosis as a reference for clinicians. We believe that such a classification would be invaluable when expanding/narrowing one’s differential diagnosis in headache.

## 2. Materials and Methods

Our study was conducted in two phases: (1) the differential diagnosis generation phase and (2) the set/subset generation phase. The differential diagnosis generation phase consisted of the building of DiffNet; we will review it here [5]. The set/subset generation phase involved determining the set/subset relationships between differential diagnosis groupings.

### 2.1. The Building of DiffNet, the Differential Diagnosis Generator

DiffNet reinterprets ICHD3 as a graph in the following fashion: each ICHD3 diagnosis is considered a “node”; if one ICHD3 diagnosis references another ICHD3 diagnosis anywhere in the criterion or comment section, then an “edge” exists between the two. Furthermore, an edge exists between two ICHD3 diagnoses if they are connected through the ICHD3 hierarchy.

For example, 6.1 and 6.1.1 are connected, since the latter is a sub-section diagnosis of the former. However, 6.1.1.1 and 6.1.1.2 are not necessarily connected as an edge, as they are on the same level. (On the other hand, 6.1 and 6.1.1.1 are connected, although they are once removed.) This ICHD3 graph is undirected and contains 387 nodes (due to diseases with duplicate diagnostic codes, for example, “complications of migraine”, 1.4 and A1.4). An editorial choice was made by DiffNet to consider duplicated diagnosis the same as long as they are called by the same name; for example, 1.4 and A1.4 are considered the same entity.

ICHD3 labels cross-references between headache disorders through its numerical diagnostic codes. We exploited this consistency computationally to generate the above graphical representation without the use of natural language-processing algorithms. To make sure no parsing errors exist, the data were manually reviewed and verified by the author.

DiffNet then operates in the following fashion to generate differential diagnosis: when given an ICHD3 diagnosis code as an input node, DiffNet look for all “first-degree neighbors” of that diagnostic code. First-degree neighbor is graph theoretic parlance for all nodes connected to a given node via an edge. These “first degree neighbors” are output to the user as differential diagnosis. (DiffNet also provides “second degree neighbors” as well to describe less likely differential diagnoses. We ignore these in this project.)

Consider “chronic tension-type headache” as an example—the “first degree neighbors” of this node/ICHD3 diagnosis are the following:Tension-type headacheChronic tension-type headache associated with pericranial tendernessHeadache attributed to psychiatric disorderMigraine without auraNitric oxide (NO)-donor-induced headacheProbable chronic tension-type headacheMedication-overuse headache (MOH)Chronic migraineFrequent episodic tension-type headacheHeadache attributed to idiopathic intracranial hypertension (IIH)Chronic tension-type headache not associated with pericranial tendernessNew daily persistent headache (NDPH)

In other words, the above are ICHD3 diagnoses that are either mentioned in the criterion/comment section—such as “medication-overuse headache” (8.2)—or connected via ICHD3 hierarchy—such as, “chronic tension-type headache not associated with pericranial tenderness” (2.3.1). Note that edges that are established elsewhere in ICHD3 were also included in the differential. For example, “headache attributed to idiopathic intracranial hypertension” was included, since it was mentioned not in the chronic tension-type headache section but rather in 7.1.1 as a potential differential: “7.1.1 Headache attributed to idiopathic intracranial hypertension may mimic the primary headaches, especially 1.3 Chronic migraine and 2.3 Chronic tension-type headache; on the other hand, these disorders commonly coexist with IIH.” (page 101 of ICHD3) [3].

The building of DiffNet was accomplished through Python with NetworkX library; C# was used as an interface. The algorithm is freely distributed through user registration by Rutgers University at http://license.rutgers.edu/technologies/2019-070_diffnet-head-ache-differential-diagnosis-software (accessed on 26 October 2021). (To use the algorithm, simply enter input diagnosis as lower case.) The source code is proprietary to Rutgers University but could be opened to limited disclosure with inquiry to the author. The parsing of ICHD3 for DiffNet was accomplished through the Haskell programming language.

### 2.2. Determining Set/Subset Relationships

Once DiffNet was generated, each of the ICHD3 diagnoses codes were algorithmically entered into DiffNet to generate a list of differential. We call the set of differential diagnosis generated by a specific ICHD3 diagnostic node the differential diagnosis set *induced* by that specific diagnostic code. For example, in our example above, the set of differentials induced by “chronic tension-type headache” is the first-degree neighbors generated by DiffNet.

Once sets of differential diagnoses induced by each of the ICHD3 diagnostic codes were obtained, we then algorithmically obtained all possible *proper subset* relationships between differential diagnosis sets.

Recall that subset and proper subsets are mathematically defined as the following:

Let A and B be sets, B is a subset of A if and only if every member of B is a member of A. We call A the super set of B. Furthermore, B is a proper subset of A if it is strictly contained in A [6,7,8].

Consider the following example on three sets. The differential diagnosis set induced by “migraine with aura” is:Familial hemiplegic migraine type 3 (FHM3)Typical aura with headacheFamilial hemiplegic migraine type 1 (FHM1)Pure menstrual migraine with auraProbable migraineChronic migraine (alternative criteria)Headache attributed to moyamoya angiopathy (MMA)Episodic syndromes that may be associated with migraineSporadic hemiplegic migraine (SHM)Benign paroxysmal torticollisHeadache attributed to mitochondrial encephalopathy lactic acidosis and stroke-like episodes (MELAS)Angiography headacheHeadache attributed to an intracranial endarterial procedureFamilial hemiplegic migraine (FHM)Triptan-overuse headacheMenstrually related migraine with auraFamilial hemiplegic migraine type 2 (FHM2)Headache attributed to other chronic intracranial vasculopathyMigraine aura statusHeadache attributed to transient ischaemic attack (TIA)Status migrainosusVestibular migraineTypical aura without headacheMigraine with typical auraMigraine aura-triggered seizureFamilial hemiplegic migraine other lociRetinal migraineVisual snowMigraine with brainstem auraInfantile colicChronic migraineHemiplegic migrainePersistent aura without infarctionHeadache attributed to cerebral autosomal dominant arteriopathy with subcortical infarcts and leukoencephalopathy (CADASIL)Probable migraine with auraHeadache attributed to arteriovenous malformation (AVM)Migrainous infarctionNon-menstrual migraine with auraMigraine without auraHeadache attributed to cerebral venous thrombosis (CVT)MigraineMigraine with aura

The differential diagnosis set induced by “familial hemiplegic migraine (FHM)” is:Familial hemiplegic migraine type 3 (FHM3)Familial hemiplegic migraine type 1 (FHM1)Hemiplegic migraineFamilial hemiplegic migraine (FHM)Sporadic hemiplegic migraine (SHM)Familial hemiplegic migraine other lociFamilial hemiplegic migraine type 2 (FHM2)MigraineMigraine with aura

Furthermore, differential diagnoses induced by “familial hemiplegic migraine type 3 (FHM3)” are as follows:Familial hemiplegic migraine type 3 (FHM3)MigraineFamilial hemiplegic migraine (FHM)Hemiplegic migraineMigraine with aura

We can write that the differential diagnosis set induced by “familial hemiplegic migraine type3” (we call this the *differential set* for short) is a proper subset of differential set induced by “familial hemiplegic migraine (FHM)”, which, in turn, occurs in a proper subset of differential set induced by “migraine with aura”. Symbolically, we can write the following: differential diagnosis set induced by “familial hemiplegic migraine type 3” < differential set induced by “familial hemiplegic migraine (FHM)” < differential set induced by “migraine with aura”.

All such set/subset relationships between differential diagnosis sets in the ICHD3 were determined and recorded.

We then algorithmically determined the minimum set of ICHD3 whose differential diagnoses encompass the totality of ICHD3. We call these sets the *lower-sets*. We further analyzed the resultant set/subset relationships by comparing them with the ICHD3 categories.

The set/subset phase of the project was accomplished using Haskell programming language. A graphical representation of this subset relationship is presented in Appendix A and was generated using Gephi. Appendix A is available as part of the Appendix A. Codes are available upon request from the author with permission from Rutgers University.

## 3. Results

A total of 1368 edges and 387 nodes were generated by DiffNet. There are 584 proper subset relationships. The differentials encompass all of the ICHD3 diagnoses. Of these, all 14 first digit levels of the ICHD3 diagnostic criteria are present.

Appendix A displays the 93 lower-sets, of these, 51 contains subsets. (Due to their size, Appendix A are included as Appendix A.) There are 42 differential sets (we call them *singletons*) that are neither subset nor a superset of another set. We display these in Appendix A. The complete lists of all set/subset relationships for sets in Appendix A are displayed in Appendix A, hierarchically, similar to ICHD3. (The 42 sets in Appendix A have no set/subset relationship and, therefore, cannot be displayed hierarchically.) We will reference each diagnosis according to their section and subsection numbers based on Appendix A. For example, “visual snow” is “31 v”. We will refer to singletons by their name.

The 14 first digit levels of ICHD3 criteria are represented by the following differential groups:Migraine is represented by sets 26, 31, 32, 33, 35.Tension-type headaches is represented by sets 6, 9, 29, 48, 49, 50.Trigeminal autonomic cephalalgias is represented by sets 5, 7, 25, 46, 47.Other primary headache disorders are represented by sets 8, 28, 37, 42, 43, 44, 45.Headache attributed to trauma or injury to the head and/or neck is represented by sets 1, 22, 41.Headache attributed to cranial or cervical vascular disorder is represented by sets 2, 11, 17, 20, 39, 40.Headache attributed to non-vascular intracranial disorder is represented by sets 13, 14, 16, 18, 21.Headache attributed to a substance or its withdrawal is represented by sets 3, 10, 23, 27, 30, 34.Headache attributed to infection is represented by sets 15.Headache attributed to disorder of homeostasis is represented by sets 12.Headache or facial pain attributed to disorders of the cranium, neck, eyes, ears, nose, sinuses, teeth, mouth or other facial or cervical structure is represented by sets 4, 24.Headache attributed to psychiatric disorder is represented by sets 19.Painful lesions of the cranial nerves and other facial pain is represented by sets 38, 51.Other headache disorders are represented by sets 36.

We present the set/subset relationship graphically with arrows pointing from the super set to its subset for visualization in Appendix A. (See Appendix A). We excluded the 42 singletons from this graph, as these datapoints contain no arrow.

### 3.1. Difference and Similarities with ICHD3

A number of super sets mirrors their ICHD3 counterparts, suggesting that a number of ICHD3 diagnoses hierarchy do contain value for differential diagnosis generation:Headache attributed to homeostasis (super set 12)Headache attributed to infection (super set 15)Headache attributed to non-vascular intracranial disorder (super set 18)Headache attributed to psychiatric disorder (super set 19)Headache or face pain attributed to disorder of the cranium, neck, eyes, ears, nose, sinuses, teeth, mouth, or other facial or cervical structure (super set 24)Painful lesions of the cranial nerves and other facial pain (super set 38)Trigeminal neuralgia (super set 51)

Despite these similarities, the hierarchy generated by this project is in general “wider” yet “shallower” hierarchically than the ICHD3. For example, the maximum depth of ICHD3 is five levels (cf ICHD3 8.2.3.2.1 “acetylsalicyclic acid over-use headache”), whereas the differential diagnosis hierarchy is, at maximum, four levels deep. On the other hand, there are 3 parts with a total of 14 categories in ICHD3; our project requires a total of 93 differential diagnosis sets. In other words, differential diagnosis hierarchy splits the disease classification hierarchy.

### 3.2. Inversion of Orders and Violators of ICHD3 Orders

Only 12 differential diagnosis sets violate ICHD3 hierarchical order. Consider two examples of such “inversions”/”violators”: A1.2 Migraine with Aura is hierarchically not connected to A1.6.4, infantile colic, but the differential set of the latter is a subset of the former. Diagnosis code 8.1.4.2 Delayed alcohol-induced headache is under 8.1.4 alcohol-induced headache; however, the former is a superset for the latter. We present these in Appendix A. Here, superset and subset relationships are represented by the format “superset, subset”.

For the most part, these “violators” of ICHD3 hierarchy are simply minor restructuring of closely related ICHD3 diagnosis. Two of these “violators”, for example, are putting differential sets induced by “idiopathic” diagnosis under “secondary” diagnosis: for example, differential set induced by idiopathic nervus intermedius neuralgia is a subset of secondary nervus intermedius neuralgia. This simply reflects the reality of differential diagnosis generation: the differential for considering secondary cause should be “bigger” than the differential of idiopathic causes for a disease, where, presumably, one considers the latter only when a number of known causes have been ruled out.

Finally, the “lower bounds” (i.e., “meet”) of differential sets were calculated and presented in Appendix A. Appendix A are included to explore this and other theoretical implications of this project.

## 4. Discussion

The International Classification of Headache Disorders is the canonical classification and diagnostic guide in headache medicine [3]. In addition to offering a hierarchical structuring of all headache disorders, ICHD3 also contains cross-referenced comments describing the inter-connections among diagnoses. These cross-references within the ICHD3 criteria can be exploited for differential diagnosis generation in the form of DiffNet, a differential diagnosis generator published by Rutgers University.

This project seeks to answer two questions using DiffNet: (1) What are the set/subset relationships between differential diagnosis in headaches? (2) What is the minimum number of sets of differential diagnosis sets needed to cover all ICHD3 diagnosis codes? The answers to both questions offer a unique perspective on headache classifications, with implications for both clinical practice and education: (1) Since differential diagnosis comes from clinical presentations, in addressing the first question, our project can be viewed as a phenomenological re-classification of ICHD3. (2) Our project’s answer to the second questions—the 93 headache disorders the differential diagnosis sets that encompass all ICHD3—may aid trainees and clinicians when learning or applying the ICHD3.

An important caveat: when dealing with differential sets, we must remember to not think of only the disease entity the differential set is named after, but also its underlying set of differential diagnoses, which is what they are in reality.

(A note on notations: For super sets, we referenced each diagnosis sets by their section and subsection numbers based on Appendix A, an expanded version of Appendix A. For singletons, we referenced them by names based on Appendix A.)

### 4.1. Phenomenological Classification versus Pathophysiological Classification

Differential diagnoses are driven by disease phenotypes—to ask for the differentials for a specific disorder is to ask for a list of other disorders with similar presentations. The classification of differential diagnosis sets can, therefore, be considered a classification of disease phenotypes. In other words, to ask the question “What is the set/subset relationships between differentials?” is equivalent to asking for a classification of headache phenotypes.

This is evident when considering the similarities and differences between ICHD3 classification and the differential diagnosis classification offered here. Since ICHD3 is a classification based on pathophysiology, the similarities or differences between ICHD3 hierarchies and our restructuring highlights the varying presentations of disorders with similar pathophysiology.

Firstly, ICHD3 diagnoses hierarchy does often contain value for differential diagnosis generation. For example, when diagnosing a patient with “migraine with aura”, the ICHD3 allows for practitioners to effectively classify and generate a potential differential diagnosis for the specific types of migraine aura. Therefore, a number of super sets are mirrors of their ICHD3 counterparts; for example, headaches secondary to homeostasis, infections, non-vascular intracranial disorders, psychiatric disorder, facial or cervical structure, and cranial are similar in both classifications. Intuitively, the phenotypes within these sets reflect the underlying pathophysiology. For example, diseases from different homeostasis may present phenotypically similar; therefore, its diagnostic investigation (i.e., differential diagnoses) reflects its classification.

The differences between the two classifications, however, are more telling: Our differential diagnosis hierarchy splits a number of ICHD3 disease classifications, suggesting that a number of disorders with unifying pathophysiology often present differently. Consider the following non-exhaustive list of differences between ICHD3 hierarchy and the hierarchy of differential diagnosis sets:

First, a number of stroke sub-classifications in ICHD3 become disparate super sets in the differential hierarchy—the differential sets for “acute headache or facial or neck pain attributed to cervical carotid or vertebral artery dissection” (super set 2), “headache attributed to non-traumatic intracranial haemorrhage” (super set 17), “headache attributed to reversible cerebral vasoconstriction syndrome” (super set 20), “persistent headache attributed to past ischaemic stroke ”(super set 39), and “persistent headache attributed to past non-traumatic intracranial haemorrhage” (super set 40) are not subsumed under super set 11, “headache attributed to cranial and/or cervical vascular disorder” (see Appendix A). The likely explanation for this is that super set 2, 17, 20, 39, 40 are vascular disorders with symptoms that are phenotypically (although not necessarily pathophysiologically) different than the typical super set 11 presentation. Consider RCVS (super set 20); for example, the thunderclap onset of RCVS is distinct from the classical presentation of headaches from ischemic stroke (per ICHD3′s own admission, ischemic stroke headache presents in ischemic stroke in only one-third of the cases whereas the phenotype of RCVS as thunderclap made it into the criteria itself). The immediate implication of this clinical observation is that clinicians should view vascular headaches as having six different phenotypes of presentations.

Similarly, traumatic headaches are described by the sets “acute headache attributed to traumatic injury to the head” (super set 1), “headache attributed to trauma or injury to the head and/or neck” (super set 22), and “persistent headache attributed to traumatic injury to the head” (super set 41). The likely etiology for this difference is that acute versus persistence of headache post-traumatically represents two very distinct phenotypes in post-traumatic headaches. (We are not, of course, stipulating whether this is a correct/adequate classification of post-traumatic headaches, as is being questioned by a number of the recent research; we are simply stating that, according to ICHD3, the acute versus chronic phenotype appears to be salient distinction between ICHD3 subtypes of post-traumatic headaches) [9].

High or low CSF pressure headaches, although pathophysiologically linked, present differently in that their differentials form separate sets (super sets 14, 16, 21). This ought not to be a surprise to any clinicians, as elevated CSF pressure is associated with symptoms such as papilledema and vision changes, while low CSF pressure is associated symptoms such as orthostatic headache [3]. In addition to these textbook presentations, paradoxical cases exist [10].

We note that migraine is classified under various different super sets: 26, 31, 32, 33, and 35. This highlights that, phenotypically, migraine without aura presents differently to migraine with aura (namely, the aura phase). Thus, the aura phase induces very different clinical considerations in differential diagnosis.

Finally, the ICHD3 classification of “other primary headache disorders” (super set 38) contains a number of headache conditions that are so different that each becomes their own super sets of differential diagnosis. This is evidenced by the fact that super set 8, 28, 42, 43, 44, 45 ought to be included in super set 38. The likely reason for this is that although super set 38 contains these diagnoses due to ICHD3 hierarchy, these diagnoses are so disparate that they simply contain their own sets of differentials.

### 4.2. An Approach to Learning/Using ICHD3

Our project offers an answer to the question: “What is the minimum number of differential diagnosis sets that needs to be obtained in order to cover all of ICHD3 diagnosis?” We present the answer in Appendix A, as 93 differential diagnoses sets encompass all ICHD3 diagnoses. Since these lower sets allow for a complete differential diagnosis of all of ICHD3 differentials, they can be considered an efficient first point of entry for a trainee when learning the ICHD3.

Of particular interest to trainees and clinicians are the 42 differential sets that we call “singletons”—these are differential diagnoses sets that are neither a subset nor super set of another differential diagnosis sets in the ICHD3. These diagnoses appear to describe “special cases” that defy phenomenological hierarchical classification in the following fashion: (1) Some of these sets appear to exist because their headache phenotype is so stereotypical as to provide a very narrow differential. Consider, for example, the singletons MELAS and trochlear headaches—the former contains stereotypical image findings while the latter contains a stereotypical exam finding [11,12]. In other words, these differential sets are so narrow that they are almost pathognomonic (therefore, its differential set contains no subsets). (2) On the other hand, other singletons appear to stand alone because the differential for them is broad in unique combinations. In other words, these differentials are broad in such a way that no other diagnosis can completely mimic its differential. (Otherwise, they would be a super set or contain a subset.) An example is NDPH, which provides rather broad but unique differentials [13]. “Probable migraine” and “probable tension type headache”, for example, likely offer a similarly but uniquely broad differential; for the former, it is often said, only partly in jest, that “Everything is a migraine… Unless It Is not.” [14].

### 4.3. Strength and Limitations

Differential diagnosis generators can be variable in quality. Using DiffNet as a source to study differential diagnosis sets has the benefits of its being fully transparent in how differential diagnosis are generated. Specifically, there is a well-defined way of enumerating the differential for any given ICHD3 disorder under DiffNet. In addition, as DiffNet is simply a graphic representation of ICHD3, it is fully reproducible. As references between ICHD3 diagnoses are written by groups of international headache specialists, the source material for DiffNet is also of good authority. Finally, it has the benefit of being free to the public.

This project’s reliance on DiffNet and its methodology introduces limitations. Of course, cross-references in ICHD3 were not written for the purpose of differential diagnosis generation. As such, cross-references may not be clinical but pathophysiological, or, at times, could emphasize differences rather than similarities. This is an important limitation when using DiffNet for our project.

A number of limitations also exist because of the inherent assumptions of our project: Firstly, this project is based on the 3rd version of the ICHD. Therefore, future improvements in the diagnostic criteria have a downstream effect on the stability of the results of this project. For example, the classification paradigm for angiography headaches (6.7.2) had been put into question [15]. These studies would inevitably lead to a modification of headache classifications in the future and, as a result, modify our research findings. Secondly, the differential diagnoses presented in the algorithm only include disorders that are properly codified in the ICHD3. Therefore, important differential diagnoses that are not codified in the classification schema are de facto missing. Consider the differential for TIA, for example, in cases for migraine with aura—here, the ICHD3 criteria itself allow for good differentiation between this disorder and migraine with aura [16,17].

Furthermore, our prior editorial decision to include ICHD3 “alternative diagnosis” in DiffNet has a downstream effect on this project: migraine or tension type headaches are presented with multiple definitions and, as such, comes with differential sets of differential diagnosis. A future direction should be to explore the effect of reconciling these differences.

Of course, there is no reason why another differential diagnosis algorithm cannot be used to generate a differential diagnosis hierarchy using our methodology. Indeed, we encourage such an enterprise by readers to explore what would happen to differential diagnosis hierarchy if a neurology textbook, an alternative commercial project, or even an internal medicine textbook was used as input rather than DiffNet.

### 4.4. Clinical Implications of DiffNet and Future Directions

Methodologically, the ability to explore differential diagnosis as subset relationships allows for a well-structured classification of clinical reasoning in headache medicine. Such a structured view of differential diagnosis in headaches may allow for clinicians to expand/narrow their differential structurally by tracing differential diagnosis hierarchy. As such, we hope that Appendix A would become a useful resource for clinicians.

Finally, in DiffNet, all edges in the graphical representation of ICHD3 are weighted the same. Of course, the equal weighing of edges in a graph is not necessary and may not be beneficial in real-life settings. This inevitably leads to the following question: How much weight should be given to a specific diagnosis? How should these determined? For example, although an argument can be made that more weight should be given to edges that connect to “migraine without aura” as opposed to “MELAS”, in clinical settings, a characteristically abnormal MRI may bias the clinician in a completely different direction. It is the author’s speculation that the applications of novel machine learning or natural language processing techniques over structure interview data may provide the answer. For example, consider a recent study where a structured electronic diary allows for differentiation and, therefore, differentiates between “migraine” or “tension-type headache” [18]. Therefore, a future direction of our project would be the incorporation of artificial intelligence in the weighing of differential diagnoses connections.

## 5. Conclusions

We believe that our project has implications for headache classification, headache education, and as a differential diagnosis reference tool. For headache classification, our schema suggests that a future headache classification schema may consider the separation of a number of phenotypically different diseases. For headache education, we believe that education on the 93 differential diagnosis super sets would provide a more comprehensive headache curriculum. Finally, we believe the classification schema in our project would serve as an effective differential diagnosis tool for clinicians to widen or narrow differential diagnosis for real-life scenarios in headaches; indeed, Appendix A of this project is intended as such a reference guide. As a methodological exploration, we envision that the methods used in this project could be applied to other differential diagnosis generators.

## Data Availability

The data presented in this study are available in the supplementary material here.

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
