# Peer review of "The Structure and Organizations of ICHD-3 Differential Diagnoses through DiffNet: A Pilot Study"

_diagnostics, 2022, doi:10.3390/diagnostics12112589_

Round 1

Reviewer 1 Report

This is a very interesting, pioneer study which suggested a new approach for headache classification, headache education, and differential diagnosis using DiffNet, a differential diagnosis generator. The building of DiffNet was accomplished through Python with NetworkX library; C# was used as an interface The parsing of ICHD3 for DiffNet was accomplished through the Haskell programming language.

I think this programme will be very useful for neurologists and other specialists in the differential diagnosis of headache disorders. However, I have several questions:

1)      I wonder who has done this huge work?  It is very difficult to do for a neurologist without education in IT. Is it possible to do this only for one author?

2)      The Title should be more specific concerning your study, for example, “DiffNet programme in differential diagnoses of headache disorders of ICHD-3: a Pilot Study”.

3)      Keywords: Headache Classifications, Order Theory, Category Theory, Differential Diagnosis. These keywords do not describe fully this study and I suggest including more informative words for future search

4)      I suggest including a description of the building of DiffNet in the Methods of the abstract

5)      It would be better to write more about the limitations of DiffNet connected with the absence of diagnostic criteria for many neurological disorders which can influent the structure and organization of headache differential diagnoses. For example, in the case of differential diagnosis of migraine with aura and transient ischemic attack (TIA), it is necessary to include diagnosis of TIA and diagnostic criteria of TIA too since only TIA represents the most difficult and frequent disorder in the differential diagnosis of TIA and migraine with aura [ 1 ] but not other headache disorders in the list of your article.

6)      I think you should change your example of “cervicogenic headache” since this headache diagnosis is very rare and usually wrong. I suggest using tension type headache (TTH) as an example.

7)      It would be more interesting if you could provide more examples in attachment, for example, migraine without aura is the most frequent headache disorder after TTH and each neurologist should know this disorder and differential diagnosis.

8)      In general, it would be better to reorganize your schema and make a priority for the most frequent headache disorders in the differential diagnosis.   It is one of the limitations of your study.  

1.     Lebedeva ER, Gurary NM, Gilev DV, Christensen AF, Olesen J. Explicit diagnostic criteria for transient ischemic attacks to differentiate it from migraine with aura. Cephalalgia. 2018 Jul;38(8):1463-1470.

Reviewer 2 Report

The Author present an interesting report on the existing biases of the International Classification of Headache Disorders 3. We know well that this classification, whose last update is from 2018, is born in 2004 on the basis of the simple clinical experience and without including specific biomarkers both for the definition of the single types of headache and for differential diagnosis. The Author, in fact, stigmatize specifici bias that should be taken into account in the ongoing 4th revision of ICDH. The validity of the intrinsic concept of this article is high because the improvement of the differential diagnosis among primary and secondary headaches is a strategic point in the quality of education in this area of clinical medicine.

I only suggest to implement the reference section, very technical, with some suggestions already existing in current literature: 

PMID: 31910800

PMID: 32532222

PMID: 35100967

Author Response

Thank you very much for your kind comments. I have added the related literature references to the manuscript.  It is my hope that this manuscript will allow for improvement of the existing classification system.  Thank you for making this manuscript better.